# School-based universal mental health promotion intervention for adolescents in Vietnam: Two-arm, parallel, controlled trial

Thach Duc Tran[1] , Huong Nguyen[2], Ian Shochet[3], Nga Nguyen[2], Nga La[2], Astrid Wurfl[3], Jayne Orr, Hau Nguyen[1], Ruby Stocker[1] and Jane Fisher[1]

[1]Global and Women's Health, Public Health and Preventive Medicine, Monash University, Melbourne, VIC, Australia; [2]Department of Health Promotion, Hanoi University of Public Health, Hanoi, Vietnam and [3]School of Psychology and Counselling, Queensland University of Technology, Brisbane, QLD, Australia

## Research Article

**Keywords:**
depression; universal intervention; adolescents; school; low- and middle-income countries

**Corresponding author:**
Thach Duc Tran;
Email: Thach.tran@monash.edu

## Abstract

The aim of this study was to evaluate the effects of Happy House, a universal school-based programme, in reducing adolescents' depressive symptoms and improving their mental well-being, coping self-efficacy and school connectedness. This was a school-based, two-arm parallel controlled trial. Depressive symptoms were measured using the Centre for Epidemiologic Studies Depression Scale. Data were collected at recruitment, and at 2 weeks and 6 months post-intervention. Mixed-effect models were conducted to estimate the effects of the intervention on the outcomes. A total of 1,084 students were recruited. At 2 weeks post-intervention, the effect size on depressive symptoms was 0.11 ($p = 0.011$) and the odds of having clinically significant depressive symptoms were lower in the intervention compared to the control (0.56, $p = 0.027$). Both of these were no longer significant at 6 months post-intervention. Psychological well-being mean scores in the intervention were significantly higher than in the control at 2 weeks post-intervention (effect size 0.13). Coping self-efficacy mean scores were significantly higher in the intervention group at both 2-week and 6-month post-intervention (effect sizes from 0.17 to 0.26). Data support the potential of Happy House to reduce the prevalence of adolescent mental health problems and to promote positive mental health in the school context in Vietnam.

## Impact statement

In high-income settings, school-based universal mental health promotion programmes have been proven to be effective at improving adolescents' mental health and well-being, including preventing and reducing the prevalence of depression in a large number of studies. Very few similar studies have been conducted in low- and middle-income countries. Among those, four studies found that school-based universal mental health promotion interventions had positive effects on students' depression and anxiety. This universal school-based mental health programme can improve adolescents' psychological well-being, coping self-efficacy and anger management, in addition to reducing depressive symptoms. The effects were observed not only immediately after the intervention, but some also at the 6 months follow-up. This programme, if integrated into the existing school curriculum in Vietnam and other low-and middle-income countries, might have potential long-term benefits for adolescents' mental health and their physical health, academic performance and productivity in adulthood.

## Introduction

Depression among adolescents is a major public health problem worldwide as it is the fourth most frequent cause of adolescent disability-adjusted life years (GBD 2019 Diseases and Injuries Collaborators, 2020). Universal mental health promotion interventions are delivered to a broad population and aim to strengthen positive aspects of mental health, regardless of prior mental health status or risk factors. The school setting is well-suited and has necessary resources for sustainable large-scale roll-out of universal mental health programmes for adolescents, which use health education approaches and do not require professionals (Regan et al., 2020). In high-income countries, several school-based universal mental health promotion programmes were found to be effective at improving adolescents' mental health and well-being, including preventing and reducing the prevalence of depression (Arora et al., 2019; Feiss et al., 2019).

As school enrolment rates among children and adolescents increase rapidly in low- and middle-income countries (LMIC) (UNESCO, 2015), there is increased recognition of the potential of universal school-based mental health programmes in preventing adolescent mental

health problems. In a recent systematic review (Caldwell et al., 2019) of school-based universal interventions targeting depression and anxiety, only five (Bonhauser et al., 2005; Gallegos, 2008; Rivet-Duval et al., 2011; Araya et al., 2013; Velásquez et al., 2015) of 76 studies were conducted in LMIC. Four (Bonhauser et al., 2005; Gallegos, 2008; Rivet-Duval et al., 2011; Araya et al., 2013) of the five studies reported positive effects on students' mental health outcomes. Another systematic review of universal school-based mental health programmes in LMIC found that nine of the 12 included studies reported significant effects on depression and/or anxiety at post-evaluation (Bradshaw et al., 2021). A number of recent school-based trials aimed to address adolescent mental health problems in LMIC (Michelson et al., 2020; Malik et al., 2021; Osborn et al., 2021). However, these studies all targeted adolescents with existing clinically significant symptoms of mental health problems; none investigated a universal intervention.

Most universal mental health programmes, particularly school-based programmes, have been developed in high-income countries (Fazel et al., 2014a). Adaptation of evidence-based programmes developed in high-income countries for use in resource-constrained settings is a cost-effective and widely-used approach (Chowdhary et al., 2014). The evidence on the effectiveness of a number of adopted mental health interventions in new settings is positive (Benish et al., 2011; Chowdhary et al., 2014). However, some trials of programmes adopted and implemented in other countries had smaller effect sizes than those implemented within the country of development (Wigelsworth et al., 2016). The complexity of international transferability of mental health programmes across countries and cultures is significant. A rigorous process from initial feasibility and acceptability assessment, cultural adaptation and testing of the adapted programme are crucial to ensure adequate effectiveness, as if implemented in the original setting (Bernal et al., 2009).

The Resourceful Adolescent Program (RAP) (Shochet and Wurfl, 2015), a universal school-based programme developed in Australia, aims to build psychological resilience and enhance the coping resources among adolescents to prevent depression and promote positive mental health. The RAP is a commercialised programme. The RAP is delivered in facilitated groups and comprises activities informed by cognitive behaviour therapy (CBT) (Beck et al., 1979) and interpersonal psychotherapy (IPT) (Mufson et al., 1993). Aspects of CBT are incorporated into RAP to support adolescents to improve their ability to keep calm in stressful circumstances, engage in positive self-talk, and strengthen problem solving capacity, all to ultimately enhance self- and affect regulation. In addition, IPT is included to assist adolescents to build and maintain support networks, and to encourage the use of perspective and empathy to reduce interpersonal conflict.

The effectiveness of RAP has been reported in three studies, particularly from English speaking, high-income countries. In Australia, one trial found that two intervention groups, RAP-A (RAP for adolescents) and RAP-F (RAP-A plus a programme for parents), had significantly larger decreases in depressive symptoms at post-intervention and 10-month follow-up, in comparison to the control group, who received the normal school curriculum (Shochet et al., 2001). Another trial compared RAP-Kiwi, the New Zealand version of RAP, to a placebo (art and craft activities without CBT) (Merry et al., 2004). To enhance the acceptability to New Zealand teenagers, the overall structure of the RAP was kept, but language, cartoons, and anecdotes were adapted in consultation with teenagers, teachers and Asian and Pacific advisers. At post-intervention, the RAP-Kiwi group had significantly larger decreases in depression scores for both the Beck Depression Inventory II (Beck et al., 1996)

and the Reynolds Adolescent Depression Scale (Reynolds, 1986). Over the 18-month follow-up period, the Reynolds Adolescent Depression Scale scores remained significantly different between the intervention and placebo groups.

The RAP has been used in 12 LMICs in the Pacific Islands, Asian, African and North American regions. However, Rivet-Duval et al.'s Mauritius-based study is the only one that has assessed the effectiveness of the RAP among secondary school students (Rivet-Duval et al., 2011). In that study, the overall structure and content of the RAP was retained after consultation with teachers about cultural relevance. The programme was delivered by teachers in the English language. The intervention group had improved self-esteem and coping skills at post-intervention and 6-month follow-up in comparison to the control group (normal curriculum only). The intervention had decreased symptoms of depression post-intervention, but not at 6-month follow-up. There is to date no study assessing the effectiveness of a translated version of RAP in an LMIC.

The importance of adolescent mental health is increasingly being recognised in Vietnam as well as in other LMICs. A recent study in Vietnam reported that up to 23% of adolescents in schools were experiencing clinically significant depressive symptoms (Thai et al., 2020). However, there are a lack of effective school-based mental health programmes that have high acceptability, feasibility and sustainability (Klasen and Crombag, 2013; Fazel et al., 2014b). As a universal strengths-based programme, RAP focuses on identifying and developing strengths for every student, rather than only targeting students who are at high risk or who have pre-existing mental health problems. By including all students, RAP enhances acceptability and minimises stigma (Shochet et al., 2001). Our team has translated RAP and adapted it for use in upper secondary schools (years 10–12) in Vietnam to create the Vietnamese version, named 'Happy House'. The aim of this study was to examine the effects of Happy House delivered in schools on the mental health of adolescents. Specifically, we tested the hypotheses that the intervention could reduce adolescents' depressive symptoms and improve their mental well-being, coping self-efficacy, and school connectedness 2 weeks after completion of the intervention and at the 6-month follow-up, when compared to adolescents receiving the regular school curriculum.

## Methods

### Study design

This study was a school-based, two-arm, parallel, controlled trial conducted in Hanoi, Vietnam (Tran et al., 2020).

### Study settings

Vietnam is located in south east Asia and is classified as a lower-middle income country, with a population of 98 million people. Hanoi is the capital city, one of the two largest cities in Vietnam. The population of eight million people in Hanoi is equally split between those living in urban and rural areas. In 2019, the school enrolment rate among school-aged children (6–18 years old) was 91.7%, an increase from 79.1% in 1999 (GSO, 2019).

### Participant recruitment

A multiple-stage method was used to select participants. First, we randomly selected two districts from a total of 12 urban districts and two from a total of 18 rural districts in Hanoi Province. Second,

two public high schools in each of the selected districts were randomly selected. In each of the selected schools, three to four grade 10 classes were randomly chosen and invited to participate in the study. Classes were only included if the school principal and class head teacher gave informed consent. In each district, we randomly assigned one school to the intervention and the other to the control arm. All participants in a single school received the same intervention. The sample selection process was performed by an independent statistician using Stata, Version 16 (StataCorp LP, College Station, Texas, USA) to create sequences of computer-generated random numbers.

All students in the selected grade 10 classes (usually aged 15–16 years) were eligible. Students received an information package consisting of two explanatory statements (one for the student and the other for their parent/s or guardian/s) and a consent form from the research team a week prior to recruitment. If a parent/guardian and student agreed for the student to participate, the parent/guardian signed the written consent form, and the student returned the form to the research team on the day of baseline survey administration. Only students who agreed and who had a parent/guardian's written consent to participate were recruited for this study.

### Intervention

Several major adaptations of RAP were made to create the Happy House programme, and these changes are described in detail elsewhere (La et al., 2022). First, Happy House was restructured from 11 45-min sessions to six 90-min sessions over 6 weeks, without reducing the content (Supplementary Tables 1 and 2). Each Happy House session spans two consecutive school periods, each lasting 45 min, with a 5-min break in between. These sessions replaced Civic Education and Ethics lessons. Second, the Vietnamese version was adapted so that it could be delivered in whole-class groups (40–45 students). The group sizes are larger than that of the original RAP (15 students). However, it is not feasible to deliver the Happy House programme in smaller groups because of constraints on facilities and human resources, and the school curriculum in Vietnam. Finally, images and videos were adapted, redrawn and remade using Vietnamese characters. The relevance, comprehensibility and acceptability of Happy House were established through two pilot tests: a four-day facilitator training course including 12 volunteer teachers and 12 Vietnamese researchers, and a pilot test with 43 participants recruited from a grade 10 class. In this study, teachers were included as facilitators, because integrating this programme into the existing school curriculum and being delivered by teachers will increase the sustainability in LMIC, where school counsellors are rarely available.

The programme has a set of illustrated participant materials and short videos, and a facilitator's manual with detailed guidelines to ensure fidelity. Fidelity was assessed using a checklist for facilitators to complete after each session, and a brief monitoring/supervision meeting between facilitators and the Vietnam Project Coordinator at the end of each intervention day. Issues that arose were discussed in these supervision meetings to reach a consensus solution. Participation rates for each session were closely monitored.

In this trial, students in the intervention group received the Happy House programme in whole-class groups, in addition to their regular school curriculum. Each Happy House session was led by a main facilitator who was a school teacher. The facilitator was assisted by a member of the Vietnamese research team. The facilitator was primarily responsible for delivering the session content, while the assistant assisted with material preparation and supported small group activities. Their roles were distinct and they were not rating students, so 'inter-rater reliability checks' between the facilitator and the assistant were neither indicated nor warranted.

The control group received only the standard academic curriculum. In Vietnam, a standard academic curriculum for each grade in high school including subjects, contents and reference books is applied for every school. The subjects the Grade 10 students have are Math, Physics, Chemistry, Biology, Technology, History, Geography, Literature, Civic Education, Secondary Language, Physical Education, and Computer Science. There are some contents related to health (in Physical Education and Biology) and interpersonal relationships (in Civic Education), but these do not include any mental health content or specific topics that presented in Happy House. The intervention group received the same standard academic curriculum.

### Procedure

The baseline survey was conducted 1 week after all participants were recruited. There were two follow-up assessments after the intervention. These were completed 2 weeks (hereafter called 'post-intervention') and 6 months post-intervention (hereafter called '6-month follow-up'). Baseline and follow-up assessments with the students were undertaken using self-completed questionnaires during a usual 45-min class session supervised by two trained data collectors from the Hanoi University of Public Health (HUPH). The instructions on how to complete the questionnaire were given orally at the beginning of the session. Students were asked to return the completed questionnaire sealed in an envelope which was provided. Data collectors were the only staff present. Students who did not want to participate and students whose parents did not grant consent for them to participate were invited to the school library to do their homework.

Data collectors and data analysts were blinded to trial arms. A code number identifying the school, but not the trial arm, was assigned for each participant. Trial arms were un-blinded after the main analyses were conducted.

### Outcomes

#### Primary outcome

Depressive symptoms were measured using the Centre for Epidemiologic Studies Depression Scale Revised (CESD-R) (Eaton et al., 2004), a 20-item scale that reflects the Diagnostic and Statistical Manual of Mental Disorders, Fourth Edition (DSM-IV) definition of Major Depressive Disorder (American Psychiatric Association, 2000). Responses to each of the items were given on a five-point Likert scale from 0 = 'Not at all or less than 1 day in the past week' to 4 = 'Nearly every day for 2 weeks'. The total scale scores ranged from 0 to 80, with higher scores indicating more depressive symptoms. A total score < 16 was used as the cut-off for 'no clinically significant depressive symptoms', as this is the widely-used cut-off score (Radloff, 1977; Eaton et al., 2004). The CESD has been validated for use among adolescents in Vietnam (Nguyen et al., 2007), but the revised version has not yet been validated in this context. We examined the construct validity of the CESD-R using the baseline data of this study. Evidence of unidimensional measurement, excellent internal consistency (Cronbach's alpha coefficient = 0.92) and measurement invariance between males and females were established (Tran et al., 2022a).

### Secondary outcomes

*Subjective mental well-being* was assessed using the Mental Health Continuum Short Form (MHC-SF) (Keyes et al., 2008). MHC-SF consists of 14 items (feelings of well-being). Each item is scored from 0 = 'Never' to 5 = 'Every day'. MHC-SF comprises three distinct subscale scores (dimensions): emotional well-being (3 items); social well-being (5 items); and psychological well-being (6 items). The emotional well-being subscale reflects positive emotional and life satisfaction. The social well-being subscale is measures self-appraisal of their positive social functioning (social coherence, social acceptance, social actualisation, social contribution, and social integration). The psychological well-being subscale reflects the extent to which an individual is realising their potential, including self-acceptance, personal growth, purpose in life, positive relations with others, autonomy, and environmental mastery. Higher dimension scores reflect better mental well-being. A validation study confirmed the hypothesised factorial structure of the three subscales and a high level of internal consistency (Cronbach's alpha coefficients of 0.81, 0.78 and 0.82 for emotional, social, and psychological well-being, respectively) among adolescents in Vietnam (Ha, 2020).

*Specific self-efficacy for coping with stress* was measured using the Coping Self-Efficacy Scale (CSES) (Chesney et al., 2006). The CSES assesses 26 behaviours that an individual might do when things aren't going well or when they are having problems. The respondent rates on an 11-point scale (0 = 'Cannot do at all' to 10 = 'Certain I can do') the extent to which they believe they could perform each behaviour. The behaviours are grouped into three categories of coping strategies (sub-scales): problem-focused (concentrating on changing the stressor itself and its physical impact), emotion-focused (managing emotional responses to the event), and social support seeking. The CSES has not been locally validated. Therefore, we examined the structural aspect of the construct validity of the CSES using this study's baseline data (Tran et al., 2022b). The original three sub-scale model was confirmed with some differences. Two items (21 and 23) did not load into any of the sub-scales and were excluded, and item two (problem-focused) and item 18 (social support) were moved to the emotion-focused subscale. In our validated version of the CSES, emotion-focused coping has nine items; problem-focused coping has the ten items; and social support coping has five items. Cronbach's alpha coefficients of the three sub-scales were at acceptable levels (emotion-focused: 0.91, problem-focused: 0.86, and social support: 0.75). Measurement invariance between males and females was supported. Sub-scale scores are the total scores of all items in each sub-scale. A higher sub-scale score indicates more self-efficacy for that coping strategy.

*School connectedness*, the bond adolescents feel towards their school, was assessed using a scale developed by The National Longitudinal Study of Adolescent Health (Sieving et al., 2001) which comprises five statements scoring from 0 = 'strongly disagree' to 4 = 'strongly agree'. The total scale score ranges from 0 to 20, where higher scores indicate higher levels of school connectedness. This scale has been used in research on adolescent's health in Vietnam (Pham, 2015).

All of these outcomes were assessed at all three timepoints (baseline, post-intervention, and 6-month follow-up).

### Baseline characteristics

We collected baseline characteristics of the participants including their age, sex (What is your sex? male or female), number of members in their household, parents' education and occupation,

and physical health (having a chronic condition such as asthma, heart disease, hepatitis, diabetes, allergies, or epilepsy or disability), using study-specific questions. The burden of academic activities was assessed using the Educational Stress Scale for Adolescents (Sun et al., 2011) which comprises 16 items about pressure from study, worry about grades, despondency, high self-expectation, and workload. Anger coping strategies were assessed using the Behavioural Anger Response Questionnaire for Children and Adolescents (BARQ-C) (Miers et al., 2007). Burden of academic activities (Pham, 2015) and anger coping strategies (Nguyen, 2010) are the major factors of depression and other mental health outcomes among student adolescents.

### Statistical analyses

#### Sample size

The number of participants was calculated using the *sampsi* command in Stata, Version 16 (StataCorp LP, College Station, Texas, USA). In each study arm, a minimum sample size of 502 adolescents was needed to detect a difference in the proportions of CESD-R score ≥ 16 of 41% (Nguyen et al., 2013) in the control arm and 31% in the Happy House intervention arm at 6-month follow-up (power of 90%, significance level of 0.05, and intra-cluster correlation of 0.01).

#### Statistical methods

The primary outcome was included in both binary (clinically significant vs. no clinically significant depressive symptoms) and continuous (CESD-R total score) formats. For each primary and secondary outcome, we performed two mixed models: Model 1 to estimate the difference in the outcome between the two arms at post-intervention and Model 2 to estimate the difference at 6-month follow-up. In these models, we controlled for all baseline characteristics presented in Table 1 (including the baseline values of the outcomes) and cluster-effects (district, school and class). The mixed-effect models incorporated random effects for district, school and class (specified as the data levels) and a fixed effect for trial arm (included in the models as an explanatory variable), which is the effect of the intervention on the outcome. Cohen's d effect sizes were calculated for significant continuous outcomes (adjusted mean difference over the pooled standard deviation). Intraclass correlation coefficients (ICC) at the class level were calculated from those mixed models. The model assumptions including multicollinearity and normal distribution of standard errors were examined. The largest variance inflation factor (VIF), which measures the correlation and strength of correlation between the predictor variables in a regression model, is less than 10 indicating little or no multicollinearity. Graphical methods (Q-Q plot) were used to examine the normality assumption of the standard errors. All analyses followed intention-to-treat principles at the individual-level using Stata, Version 16.

As outlined in the protocol (Tran et al., 2020) we also planned to conduct ancillary analyses, including subgroup analyses by attendance (participated or did not participate) and a mediation analysis, testing the effects of the intervention on the primary outcome at 6-month follow-up. In the mediation analysis, we intended to run multilevel structural equation models in MPlus Version 8 (Muthen and Muthen, 2017). The mediators that we planned to test were the secondary outcomes at 2 weeks post-intervention, as they were the intermediate outcomes that the intervention targeted to change.

In the protocol, we planned to performed multi-imputation to treat missing data. However, conducting mixed effect model

**Table 1.** Baseline characteristics of participants by study arm

| | Control group (N = 552) | Intervention group (N = 531) | p-value |
|---|---|---|---|
| *Number of clusters* | 13 | 12 | n/a |
| *Number of participants per class,* Median (min–max) | 44 (37–47) | 44 (41–48) | n/a |
| *Demographic characteristics* | | | |
| Age in years, mean (SD) | 15.3 (0.3) | 15.3 (0.3) | 0.706 |
| Females, *n* (%) | 325 (58.9) | 331 (62.3) | 0.404 |
| Urban, *n* (%) | 284 (51.5) | 256 (48.2) | 0.938 |
| Living with both biological parents, *n* (%) | 496 (89.9) | 471 (88.7) | 0.756 |
| Number of siblings, *n* (%) | | | |
| *None* | 19 (3.5) | 28 (5.3) | 0.422 |
| *One* | 318 (57.8) | 275 (52.2) | 0.338 |
| *Two or more* | 213 (38.7) | 224 (42.5) | 0.606 |
| Mother's education level, *n* (%) | | | |
| *University or above* | 220 (39.9) | 142 (26.7) | 0.437 |
| *Diploma/technical degree* | 52 (9.4) | 62 (11.7) | 0.240 |
| *High school (year 12)* | 97 (17.6) | 97 (18.3) | 0.916 |
| *Secondary school (year 9) or lower* | 93 (16.9) | 116 (21.9) | 0.502 |
| *Do not know* | 90 (16.3) | 114 (21.5) | 0.248 |
| Mother's main occupation, *n* (%) | | | |
| *Government officer* | 149 (27.0) | 103 (19.4) | 0.288 |
| *Private sector employee* | 112 (20.3) | 86 (16.2) | 0.331 |
| *Self-employed* | 147 (26.6) | 210 (39.6) | 0.191 |
| *Farmer* | 70 (12.7) | 65 (12.2) | 0.960 |
| *Not currently engaged in income-generating activity/do not know* | 74 (13.4) | 67 (12.6) | 0.700 |
| Father's education level, *n* (%) | | | |
| *University or above* | 217 (39.3) | 140 (26.4) | 0.442 |
| *Diploma/technical degree* | 35 (6.3) | 54 (10.2) | 0.106 |
| *High school (year 12)* | 106 (19.2) | 90 (17.0) | 0.717 |
| *Secondary school (year 9) or lower* | 83 (15.0) | 100 (18.8) | 0.618 |
| *Do not know* | 111 (20.1) | 147 (27.7) | 0.126 |
| Father's main occupation, *n* (%) | | | |
| *Government officer* | 133 (24.1) | 91 (17.1) | 0.282 |
| *Private sector employee* | 97 (17.6) | 81 (15.3) | 0.626 |
| *Self-employed* | 185 (33.5) | 223 (42.0) | 0.335 |
| *Farmer* | 63 (11.4) | 57 (10.7) | 0.932 |
| *Not currently engaged in income-generating activity/do not know* | 74 (13.4) | 79 (14.9) | 0.494 |
| Family has a car, *n* (%) | 189 (34.2) | 131 (24.7) | 0.360 |
| *Physical health* | | | |
| Self-reported major chronic disease and/or physical disability, *n* (%) | 58 (10.6) | 61 (11.6) | 0.757 |
| Self-reported physical health, *n* (%) | | | |
| *Very good* | 93 (16.9) | 92 (17.4) | 0.703 |
| *Good* | 193 (35.2) | 174 (32.9) | 0.935 |
| *Fair* | 250 (45.5) | 252 (47.6) | 0.860 |
| *Poor/very poor* | 13 (2.3) | 11 (2.1) | 0.899 |

*(Continued)*

**Table 1.** (*Continued*)

|  | Control group (N = 552) | Intervention group (N = 531) | p-value |
|---|---|---|---|
| *Burden of academic activities* |  |  |  |
| Educational Stress Scale score, mean (SD) | 28.1 (8.7) | 28.1 (8.0) | 0.981 |
| *Baseline data of the outcomes* |  |  |  |
| Centre for Epidemiologic Studies Depression Scale Revised score, Mean (SD) | 11.4 (12.2) | 12.0 (12.0) | 0.729 |
| Centre for Epidemiologic Studies Depression Scale Revised score ≥ 16, *n* (%) | 142 (25.8) | 133 (25.1) | 0.842 |
| Mental Health Continuum Short, Mean (SD) | 3.0 (1.1) | 2.8 (1.1) |  |
| Emotional well-being score | 10.2 (3.7) | 9.7 (3.7) | 0.120 |
| Social well-being score | 13.9 (6.4) | 13.2 (6.5) | 0.226 |
| Psychological well-being score | 17.5 (7.5) | 16.5 (6.8) | 0.075 |
| Coping self-efficacy scale (CSES) score, mean (SD) |  |  |  |
| Emotion-focused | 56.8 (19.6) | 53.3 (19.8) | 0.019 |
| Problem-focused | 68.2 (20.0) | 64.3 (19.2) | 0.036 |
| Social-support seeking | 32.0 (10.5) | 30.6 (9.9) | 0.068 |
| School connectedness scale score, mean (SD) | 18.6 (3.4) | 18.4 (3.3) | 0.724 |
| Behavioural anger response questionnaire for children and adolescents (BARQ-C), mean (SD) |  |  |  |
| Direct anger-out | 5.9 (1.7) | 5.9 (1.7) | 0.769 |
| Assertion | 8.1 (1.9) | 7.5 (2.0) | 0.006 |
| Social support-seeking | 7.5 (2.0) | 7.3 (1.8) | 0.208 |
| Distraction | 16.1 (3.2) | 16.7 (3.4) | 0.024 |
| Rumination | 7.8 (2.0) | 7.8 (1.9) | 0.634 |

analysis under the multi-imputation led to many convergence problems. The proportions of missing data were relatively small and there was no evidence of data missing not at random. Therefore, missing data were treated in several steps. For each instrument, in cases where less than 20% of items were missing, missing data were imputed using the regression imputation method to predict the item's missing value from other items of that instrument, other correlated instruments and socio-demographic characteristics. After the imputations, the remaining missing data were treated using pairwise deletion in the analyses.

### Ethics approval

This study was a part of a trial study that has been approved by Monash University Human Research Ethics Committee (Certificate Number: 21455), Melbourne, Victoria, Australia; the Institutional Review Board of the Hanoi University of Public Health (488/2019/YTCC-HD3), Hanoi, Vietnam; and Queensland University of Technology's Office of Research Ethics and Integrity (2000000087).

### Results

#### Sample

Recruitment and the baseline survey were conducted from 5th to 24th of October, 2020. The post-intervention survey was completed from 1st to 19th of December, 2020 and the 6-month follow-up data were collected between 10th and 27th of May, 2021. The 6-month follow-up data collection method was changed from a self-completed paper-based survey to an online survey (built in Qualtrics Insight Platform, Provo, UT) due to COVID-19 restrictions imposed on Hanoi during the period of data collection.

A total of 1,084 from 1,128 eligible students (96.1%) were recruited and provided baseline data. The recruitment rate was slightly higher in the intervention than the control group (Figure 1). No school or teachers selected refused to participate in this study. One participant withdrew after the baseline survey and was not included in the analyses. The number of participants with any missing data at baseline was 29 (2.7%), at post-intervention was 21 (1.9%) and at 6-month follow-up was 20 (1.9%). The missing proportions were similar in the two groups. The baseline characteristics of the groups with or without missing data of the primary outcome were not significantly different (Supplementary Table 3), suggesting that data were missing at complete random or missing at random.

Overall, all students in the intervention group attended at least five in six Happy House sessions. The number of students missing one session was 27 (5.0%).

### Baseline data

The socio-demographic characteristics of the two groups were slightly different (Table 1). In the control group, a greater proportion had parents with a higher level of education and working as government officers. The self-reported physical health and burden of academic activities were approximately similar between the two groups. There were some minimal differences in baseline data of the outcomes between the two groups including the Coping Self-Efficacy Scale subscale scores.

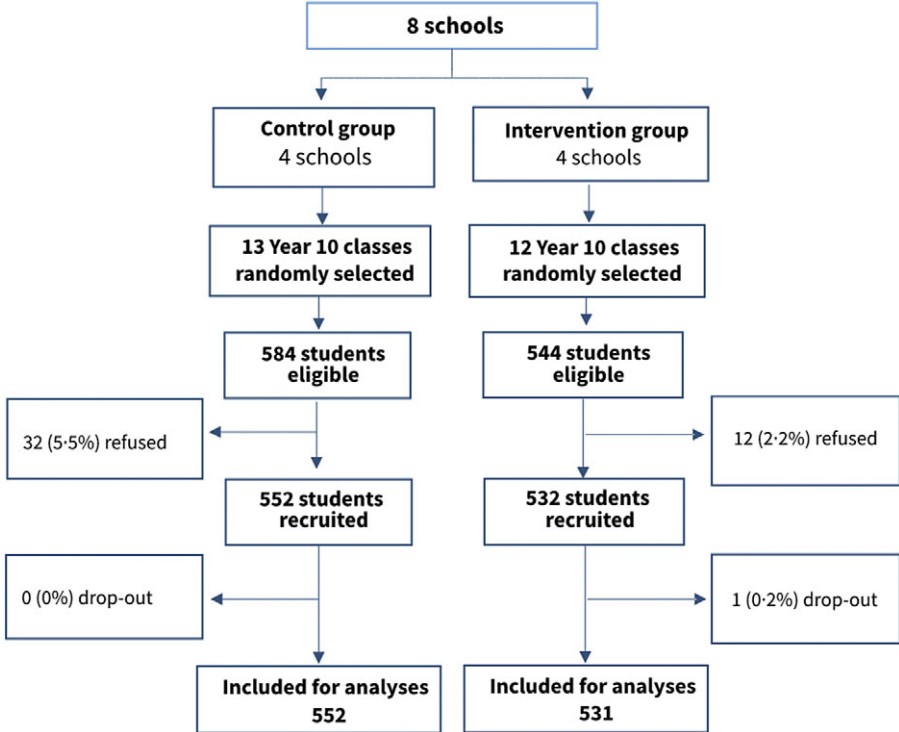

**Figure 1.** Trial profile.

### Estimation of intervention effects

The fidelity checklists were completed by the facilitators after all sessions. These indicate that 100% of activities and content were delivered as planned.

The odds of having the primary outcome (clinically significant depressive symptoms; CESD-R score ≥ 16) at post-intervention were statistically significantly lower in the intervention compared to the control group, when taking into account cluster effects and baseline data (Table 2). Similarly, the mean CESD-R score was statistically significantly lower in the intervention compared to the control group. Both of these differences became insignificant at 6-month follow-up.

There were some differences in the secondary outcomes between the two groups. The fully adjusted MHC-SF psychological well-being subscale mean differences between the intervention and control groups were significantly different at post-intervention, in favour of the intervention. All of the CSES subscale scores (with the exception of the emotion-focused subscale at 6-month follow-up) were significantly higher (more self-efficacy for coping with stress) in the intervention group at both time points. No effect of the intervention on the MHC-SF emotional and social-well-being or school connectedness was found.

VIFs in all models in Table 2 were less than 10 indicating little or no multicollinearity. The residuals of all models were approximately normally distributed.

### Ancillary analyses

We did not conduct the ancillary analysis by attendance because almost all participants in the intervention group attended all sessions. We did not perform mediation analyses because the effect of the intervention on the primary outcome at 6-month follow-up was not significant.

### Discussion

This school-based, two-arm, parallel, controlled trial addresses a major gap in evidence for the effectiveness of universal school-based mental health promotion programmes for adolescent in LMICs. Our main findings demonstrated that Happy House was effective in reducing depressive symptoms and promoting psychological well-being at post-intervention among adolescents. This study included a large sample size which was drawn from both urban and rural areas. The refusal, attrition, and missing data rates were minimal.

The prevalence of adolescents attending schools with clinically significant depressive symptoms at baseline was high but consistent with what found in previous studies in Vietnam (Thai et al., 2020). That prevalence is also similar to the results of a meta-analysis of 43 studies from many countries, using General Health Questionnaire, a common mental disorders symptom checklist (Silva et al., 2020). Symptom checklist instruments cannot provide diagnoses. However, clinically significant symptoms of mental health disorders are the target of prevention programmes. The large prevalence found in this study suggests that universal prevention programmes for adolescent mental health are warranted and in urgent need.

The proportions of students reporting clinically significant depressive symptoms increased from baseline to 2 weeks post-intervention (by 1.8%) and to 6 months follow-up (by 3.4%) in the control. It decreased from baseline to 2 weeks post-intervention (by 1.2%) and increased in 6 months follow-up (by 1.2%) in the intervention group. Concerns about potential iatrogenic harms (adverse effects) of universal school-based mental health programmes (Hsu, 1996; Foulkes and Stringaris, 2023) arise due to the possibility that providing information and self-management strategies could raise awareness, but be insufficient to some subgroups and not relevant to others. However, it's important to note

**Table 2.** Mean differences of the outcomes between trial arms at post-intervention and 6-month follow-up

| Outcome | Control group n (%) | Intervention group n (%) | Adjusted[a] odds ratio (Intervention: 1; Control: 0) | p-value |
|---|---|---|---|---|
| **Centre for Epidemiologic Studies Depression Scale revised score ≥ 16** | | | | |
| Post-intervention | 155 (28.6%) | 125 (23.9%) | 0.56 (0.36; 0.88) | 0.011 |
| Six-month follow-up | 157 (29.2%) | 138 (26.3%) | 0.75 (0.51; 1.09) | 0.132 |

| | Control group Mean (SD) | Intervention group Mean (SD) | Adjusted[a] mean difference (95%CI) (Intervention:1; Control:0) | Cohen's d effect size[b] |
|---|---|---|---|---|
| **Centre for Epidemiologic Studies Depression Scale revised score** | | | | |
| Post-intervention | 11.7 (11.5) | 11.0 (10.5) | −1.17 (−2.20; −0.14) | 0.027 | 0.11 |
| Six-month follow-up | 12.7 (12.7) | 12.3 (12.5) | −0.62 (−1.90; 0.67) | 0.349 | – |
| **Mental health continuum short form score** | | | | |
| Post-intervention | | | | |
| Emotional well-being score | 10.2 (3.8) | 9.6 (3.8) | −0.26 (−0.63; 0.10) | 0.154 | – |
| Social well-being score | 13.8 (6.5) | 13.8 (6.1) | 0.57 (−0.07; 1.21) | 0.083 | – |
| Psychological well-being score | 17.0 (7.5) | 17.1 (7.0) | 0.91 (0.06; 1.77) | 0.036 | 0.13 |
| Six-month follow-up | | | | |
| Emotional well-being score | 10.4 (3.8) | 9.8 (3.8) | −0.26 (−0.66; 0.14) | 0.203 | – |
| Social well-being score | 14.3 (6.5) | 14.2 (6.7) | 0.51 (−0.15; 1.17) | 0.132 | – |
| Psychological well-being score | 17.6 (7.2) | 17.6 (7.1) | 0.73 (−0.02; 1.47) | 0.056 | – |
| **Coping self-efficacy scale (CSES) score** | | | | |
| Post-intervention | | | | |
| Problem-focused sub-scale score | 68.7 (21.0) | 70.3 (18.6) | 4.23 (1.82; 6.66) | 0.001 | 0.21 |
| Emotion-focused sub-scale score | 58.2 (19.2) | 59.2 (17.9) | 3.18 (1.07; 5.28) | 0.003 | 0.17 |
| Social support sub-scale score | 31.7 (10.6) | 32.9 (10.2) | 2.35 (0.80; 3.90) | 0.003 | 0.23 |
| Six-month follow-up | | | | |
| Problem-focused sub-scale score | 69.5 (21.5) | 70. 6 (19.8) | 3.82 (1.00; 6.64) | 0.008 | 0.19 |
| Emotion-focused sub-scale score | 60.0 (18.9) | 60.3 (17.7) | 2.36 (−0.10; 4.83) | 0.060 | – |
| Social support sub-scale score | 30.8 (10.9) | 32.5 (10.6) | 2.82 (1.45; 4.19) | < 0.001 | 0.26 |
| **School connectedness scale score** | | | | |
| Post-intervention | 18.8 (3.7) | 18.5 (3.8) | −0.08 (−0.58; 0.41) | 0.741 | – |
| Six-month follow-up | 18.7 (3.6) | 18.5 (3.6) | −0.01 (−0.73; 0.72) | 0.985 | – |

[a]Adjusted for cluster-effects and baseline characteristics (categories) presented in Table 1.
[b]Calculated for significant continuous outcomes.

that the increase in the proportion of clinically significant depressive symptoms in the intervention group from post-intervention to the 6-month follow-up might not necessarily indicate harm caused by the programme. Several factors support this interpretation. Firstly, a significant reduction in depressive symptoms was observed immediately post-intervention. Secondly, the increase in depressive symptoms was observed in both the intervention and control groups. Finally, and the most important, this study was implemented during the COVID-19 pandemic. Throughout the 6 months follow-up period, strict restrictions were in place in Hanoi including stay-at-home lockdowns. There is evidence worldwide on the adverse impact of COVID-19 restrictions on adolescents'

mental health (Panchal et al., 2023). The pattern of changes documented in this study is likely to reflect the broader influence of the pandemic on adolescents' mental health. It suggests that programmes like this one that focus on building psychological knowledge and skills are insufficient to modify the effects of lockdowns, a special situation, on adolescent mental health. Adolescent mental health related to the pandemic in this and other resource-constrained contexts warrants further research.

The positive effect of the intervention on depressive symptoms found at 2 weeks post-intervention but not at 6-month follow-up is similar to the results of the RAP-A trial in other LMICs (Rivet-Duval et al., 2011). However, it is not consistent with the findings of two

trials of RAP-A in Australia and New Zealand, both high-income countries, which found that the effect remained 10–18 months after the intervention (Shochet et al., 2001; Merry et al., 2004). Two trials (Gallegos, 2008; Araya et al., 2013) of other school-based universal mental health interventions in LMICs followed up after a 6–12 months and assessed depressive symptoms. Both trials did not find a significant impact of the intervention on the outcome at the follow-up timepoints. There are too few studies to draw conclusions about the difference in effectiveness of interventions between high-income countries and LMICs. However, our findings and the previous literature suggest that more efforts to sustain the beneficial effects of the interventions on depressive symptoms in LMIC are warranted. Integration of regular booster sessions may be useful to maintain the positive effects of the intervention.

Compared to other analogous interventions, the magnitude of the effects of Happy House on depressive symptoms (Cohen's d of 0.11) has effect sizes smaller than some but higher than the others. In LMIC, Rivet-Duval et al. (2011) found a Cohen's $d$ effect size on post-intervention depressive symptoms of 0.32, Gallegos (2008) reported an effect size of 0.28 on post-intervention depressive symptoms and 0.10 at 6 month follow-up, and Fuspita et al. (2018) found an effect size of on post-intervention depressive symptoms of 1.57, but others did not find any significant effects (Conley et al., 2015; Velásquez et al., 2015; Bradshaw et al., 2021). Conley et al. (2015) conducted a meta-analysis of 103 universal mental health prevention programmes for higher education students and found a pooled effect size of 0.2 for depressive symptoms and the effect size of our study is higher than 29 programmes (28%). Dray et al. (2017) found a pooled effect size of 0.08 of 49 universal school-based resilience-focused interventions for children and adolescents' depressive symptoms. Tanner-Smith et al.'s review of 74 meta-analyses to assess the effects of universal mental health promotion and prevention programmes for school-age youth provides the mean effect size distributions which are the appropriate evidence-based standards for judging the relative effects of universal prevention programmes for adolescents (Tanner-Smith et al., 2018). For the interventions targeting internalising problems (anxiety, depression, or other mental health conditions), the 25[th], 50[th], and 75th percentile effect sizes are 0.10, 0.18 and 0.27, respectively. Therefore, the effect size of our intervention is larger than 25% of other universal prevention programmes. Overall, universal school-based interventions targeting adolescents' mental health have relatively small effect sizes per participant. However, universal interventions like Happy House can reach a large number of people. Therefore, even with a small change for an individual, universal interventions can have large potential for populations.

Our study is the first to date in an LMIC to assess the impact of a universal school-based mental health programme on subjective mental well-being. Mental well-being is related to, but different from, mental illness, such as depression (Keyes, 2005). Among the three dimensions of mental well-being, we found that Happy House improves psychological well-being, but not emotional or social well-being. The higher psychological well-being people have, the more they like most parts of themselves (self-acceptance), have warm and trusting relationships (positive relations with others), see themselves developing into better people (personal growth), have a direction in life (purpose in life), are able to shape their environment to satisfy their needs (environmental mastery), and have a degree of self-determination (autonomy). Through the RAP principles of building adolescents' resilience and focusing on their strengths, Happy House appears to impact psychological well-

being directly. Emotional well-being is the individual's satisfaction with life overall, and social well-being consists of social coherence, social acceptance, social actualisation, social contribution, and social integration. The intervention primarily focuses on improving relationships and reducing conflict; a much narrower focus than what emotional and social well-being covers. In this study, all three dimensions of mental well-being were measured as part of the well-being assessment. If emotional and social well-being were improved, it may have been interesting to observe that Happy House could have impacts outside the aim. However, it is not surprising that we found that Happy House improves only psychological well-being but not the other two positive mental health domains significantly.

In the CBT components of Happy House, adolescents are encouraged to develop a variety of skills to manage stressful and difficult circumstances, including maintaining positive self-talk, keeping calm (self-regulation and self-relaxation), problem-solving (defining problems, considering solutions, using a step-by-step approach for carrying out and evaluating the solution), and support networks (encouraging seeking help when necessary to maintain their emotional well-being). In this study, the positive effects of Happy House on all three coping strategies (problem-focused, emotion-focused, and social support seeking) were demonstrated. These findings are consistent with the RAP trial in Mauritius in which the intervention improved coping skills at postintervention and 6-month follow-up (Rivet-Duval et al., 2011). The improvements in adequate coping skills were reported in some other universal school-based programmes in lower-middle-income countries (Barry et al., 2013). The evidence, all together, strongly support that adolescents' coping skills can be improved through school-based universal programmes.

The matter of employing p-value adjustments in trials involving multiple outcome measures is a subject of ongoing debate. On one hand, there are recommendations advocating for p-value adjustments due to the increased risk of encountering at least one statistically significant test result by chance alone, leading to potential misidentification of differences (Bland and Altman, 1995). Conversely, adjustments can diminish the likelihood of committing type I errors (i.e., introducing ineffective treatments) but can thereby elevate the possibility of type II errors (i.e., overlooking effective treatments), an outcome equally significant to type I errors (Rothman, 1990; Feise, 2002). In the context of this study, p-value adjustments were not taken into consideration prior to the analyses (as outlined in the protocol). However, upon applying p-value adjustments to the secondary outcomes via the Bonferroni correction formula (Hsu, 1996), the strictest method, it becomes evident that the three Coping Self-Efficacy Scale subscales exhibited significant improvement post-intervention and at the 6-month follow-up, except for the Emotion-focused and Problem-focused sub-scale scores at the 6-month follow-up.

One of the adaptations of the RAP in Vietnam is increasing the group size from up to 16 adolescents in the original version to encompass the entire class (approximately 40 to 45 students), The adaption aims to enhance acceptability and feasibility when scaling up. During the pilot, we encountered several challenges related to the larger group sizes. Firstly, ensuring that the majority of students grasp the session's content adequately poses a difficulty for the facilitator. Secondly, engaging all students in discussions becomes more complex. Finally, many activities are more difficult to execute and manage within a larger group, sometimes leading to disorder and disruptions to nearby classes. To address these challenges, we formulated strategies including adding an assistant to support the

facilitator in each session and occasionally dividing the class into smaller groups for discussions and activities when necessary.

We acknowledge some limitations of this study. First, we could not randomise the classes because multiple classes per school were involved in this study. If students in the same school (but different classes) received different types of intervention, it would have increased the risk of contamination significantly. We acknowledge that the school characteristics (including teachers and students within the school) could be confounding factors that led to biases in this study. Second, we did not pair schools between intervention and control groups due to the lack of necessary data that could help balance the baseline characteristics between the two study arms. Third, the main outcome measure is not a diagnostic instrument. However, the CESD-R is the one of the most widely accepted depressive symptom checklists in research and clinical practice. This scale covers all symptoms in the DSM-IV's diagnostic criteria for Major Depressive Disorder. Fourth, all our data were self-reported, which may have introduced response biases. Fifth, fidelity in this study was self-assessed by the facilitators. There was no assessment on the delivery quality and engagement conducted. We planned for this assessment to be conducted by research team members from Australia. However, COVID-19 restrictions during the course of intervention prevented team members travelling internationally. Sixth, this study was implemented during the COVID-19 pandemic. Nevertheless, we are confident that COVID-19 pandemic did not cause any significant bias in this study because both study arms experienced the same COVID-19 restrictions during the study period. Finally, we had hoped that our study would provide information about the mechanisms of the effect of the intervention on the main outcome. However, the intervention only had an effect on the main outcome at post-intervention, the same timepoint the potential mediators were measured. Therefore, as the criterion of time order was not met, we were prevented from conducting these ancillary analyses.

## Implications and conclusions

This study demonstrates the positive impact of Happy House on adolescents' depressive symptoms, psychological well-being, and stress coping strategies. This trial had high recruitment and participation rates and no adverse effects were found. The findings of this study can be generalised to other areas in Vietnam and other countries with similar social, economic and cultural environments.

We suggest that further adaptation of this programme to improve long-term effects is warranted. This programme, if integrated into the existing school curriculum with regular booster sessions, might have potential wider long-term benefits. Effective problem solving and emotion regulation strategies for adolescents are all potentially beneficial to not only their mental health but also their physical health, academic performance and productivity in adulthood. A retest of the adapted version is also necessary to confirm the long-term effects before scaling up can be considered. In addition, a barrier for the scale up of RAP in Vietnam and other LMICs is that RAP is commercialised. It is free for research but has a charge if it is being implemented. We are conducting an economic analysis alongside this trial and the results will be published in a separate paper. That economic analysis will provide necessary information for policy makers and programmers to make decisions on taking this programme to scale.

In conclusion, this study strongly supports that Happy House has great potential to be integrated into the existing school curriculum to reduce the prevalence of adolescent mental health problems, and to promote positive mental health in Vietnam.

**Open peer review.** To view the open peer review materials for this article, please visit http://doi.org/10.1017/gmh.2023.66.

**Supplementary material.** The supplementary material for this article can be found at https://doi.org/10.1017/gmh.2023.66.

**Data availability statement.** The data that support the findings of this study are available on request from the corresponding author, T.D.T. The data are not publicly available due to their containing information that could compromise the privacy of research participants.

**Acknowledgements.** The authors are especially grateful to the adolescents who contributed their experiences to this research.

**Author contribution.** T.D.T., Hu.N., I.S., and J.F. designed this study. All authors conducted this study. H.N. managed the data. T.D.T. and H.N. conducted the statistical analysis. T.D.T. wrote the draft of this paper. All authors provided the interpretation of results and critically reviewed the draft. All authors reviewed and agreed on the content of the final submitted version. T.D.T., Hu.N., and J.F. have accessed and verified the data. T.D.T. and J.F. were responsible for the decision to submit the manuscript.

**Financial support.** This work was supported by the Australian National Health and Medical Research Council (GNT1158429), and the Vietnam National Foundation for Science and Technology Development (NHMRC.108.01–2018.02). T.D.T. is supported by a Monash Strategic Bridging Fellowship; J.F. is supported by a Finkel Professorial Fellowship, which is funded by Finkel Foundation. The funders had no role in study design; in collection, analysis and interpretation of data; in the writing of the report; and in the decision to submit the paper for publication.

**Competing interest.** The authors declare none.

**Ethics standard.** This study is part of a larger trial which has been granted ethical approval by Monash University Human Research Ethics Committee, Melbourne, Australia (approval number: 21455), the Institutional Review Board of the Hanoi University of Public Health, Hanoi, Vietnam (Certificate Number: 488/2019/YTCC-HD3), and Queensland University of Technology's Office of Research Ethics and Integrity, Brisbane, Australia (2000000087).

**Trial registration number.** Registered with the Australian New Zealand Clinical Trials Registry, registration number: ACTRN12620000088943 (3/2/2020). WHO Universal Trial Number: U1111–1246-4,079.

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
