## [Reviewer Report]

Dear Professor Gary Belkin,

We are pleased to submit our manuscript ‘School-based universal mental health promotion intervention for adolescents in Vietnam: two-arm, parallel, controlled trial’ to be considered for publication in The Cambridge Prisms: Global Mental Health. 

The importance of promoting adolescent mental health is becoming increasingly recognised in preventing mental health problems in both adolescence and adulthood. In high-income countries, school-based universal mental health promotion programs have been found to be effective at improving adolescents’ mental health and well-being. However, few similar studies have been conducted in low- and middle-income countries. The aim of our study was to examine the effects of the Happy House, a school-based universal program, on the mental health and wellbeing of adolescents in Vietnam. 

We conducted a school-based, cluster-randomised controlled trial in Hanoi, Vietnam, with 1084 Year 10 students from eight schools (four intervention; four control). The intervention group participated in six weekly 90-minute group sessions of Happy House, and the control group received the usual curriculum. Data were collected via paper-based and online surveys at baseline, two weeks and six months post-intervention. We found that students in the intervention group had reduced depressive symptoms and improved psychological well-being at two weeks post-intervention, and improved coping self-efficacy at two weeks and six months post-intervention. 

We believe that Happy House provides an excellent opportunity to be sustainably integrated into the existing school curriculum in Vietnam and other similar countries, to prevent adolescent mental health problems and promote resilience and well-being. We are grateful for your consideration of this manuscript, which we believe would be of interest to readers of The Cambridge Prisms: Global Mental Health. 

Yours sincerely,

Thach Tran, PhD, BEc, MIRB, MSc

Senior Research Fellow

Global and Women’s Health

School of Public Health and Preventive Medicine

Monash University

---

## [Reviewer Report]

This is a large randomized controlled trial of a “universal” school-based intervention to reduce depressive symptoms among over 1000 adolescents. The paper is very well written, with really interesting and potentially impactful results. I only have a few comments that might help improve it.

--It is typicial for RCT testing the efficacy of universal school-based interventions to pair schools having similar demographic characteristics, and randomize one school by pair to the intervention (e.g., Ohye et al. 2020). This greatly limits the risk for group differences at baseline.

--Relatedly, there are a couple of group differences with regards to demograpics at baseline. I believe the baseline differences in psychological outcomes are already taken care of by xtmixed, but I wonder whether the authors considered running sensitivity analyses controlling for baseline differences.

--Because of the multiple comparisons, the authors might want to consider lowering the p-value retained for the statistical significance of their analyses, especially for the secondary outcomes.

--Finally, regarding the mixed model, it is unclear whether the authors ran for each outcome, a first model with two timepoints (baseline + endpoint), and a second model with three timepoints (baseline + endpoint + follow-up) or two timepoints (baseline + follow-up). In fact, I assumed that time was treated as a within subject factor, but this is not clearly stated.

--Minor points:

-In the abstract, it is unclear whether the efficacy data refers to the posttreatment or f/u timepoint. Relatedly, the authors might want to report the outcome data for both the endpoint and f/u timepoints.

-Bar graphs may not be the best way to represent the data.

---

## [Reviewer Report]

In this well-written and well-executed study, the authors present the findings from the delivery of ‘Happy House’, which is universal school-based program (based on the Resourceful Adolescent Program) aimed at reducing depressive symptoms and improving mental health well-being in adolescents in Vietnam. This study is among few universal mental health and well-being interventions that have been delivered in low-and middle-income settings and reports to be the first to explore the effects on subjective well-being, specifically. The authors are to be commended for the successful and careful delivery of this intervention during COVID-19 and for the good outcomes – especially those related to well-being and coping.

One of the key differences reported by the authors, between the delivery of this intervention in the Vietnamese schooling context versus elsewhere in high-income settings is that classroom sizes were a lot bigger here. Are the authors able to add to the discussion perhaps more of their observations and experience as to what challenges, if any, they experienced with delivery to a large class? Also, how long are class times at the schools and with the 90 minutes sessions, were these over two lessons? Which lessons in the standard curriculum were substituted for the intervention sessions, and was this substitution the same across schools?

The intervention was delivered to year 10 learners (~15 years old) at schools in 2 urban districts and 2 rural districts in Hanoi Province. This meant 4 schools received the intervention and 4 schools received only the standard curriculum which has no specific mental health focus. Given the lack of mental health content provided through the school curriculum and given that not all year 10 classes at each of the schools allocated to the intervention group were included to receive the intervention – I wondered perhaps why the researchers did not opt for wait-list control? Same too for those who were allocated to the control schools. I mention this also given the increase depressive symptoms at the control schools at both follow up times. Did the authors experience any school concerns or school related issues with not providing the intervention to all learners in all classes?

There is much conversation in the universal and school-based mental health literature about the potential harms associated with school-based interventions. Given the slight increase in depressive symptoms at 6 months follow up in the intervention group, could the authors perhaps consider reflecting on in the discussion about potential harms and whether some aspects of this intervention could increase distress or clinical symptoms? Or perhaps to suggest in the limitations that exploring this is important in future studies?

Below are a few minor comments for the authors to consider.

Minor comments

• Please consider adding in the methods whether the trial is reported in accordance with the CONSORT guidelines?

• The authors write, “At post-intervention, the RAPKiwi group had significantly larger decreases in depression scores for both the Beck Depression Inventory II and the Reynolds Adolescent Depression Scale”. Please consider adding the references for these scales references in brackets.

• The authors collected data on other chronic conditions as well - please consider including a brief description of which chronic conditions these were?

• How many facilitators were present in each class during delivery? Were inter-rater reliability checks done if >1?